# The Lidargrammetric Model Deformation Method for Altimetric UAV-ALS Data Enhancement

**Antoni Rzonca ***[iD] **and Mariusz Twardowski** [iD]

Department of Photogrammetry, Remote Sensing and Spatial Engineering, The Faculty of Geo-Data Science, Geodesy and Environmental Engineering, AGH University of Science and Technology, al. Mickiewicza 30, 30-059 Krakow, Poland
* Correspondence: arz@agh.edu.pl; Tel.: +48-12-617-39-93

**Abstract:** The altimetric accuracy of aerial laser scanning (ALS) data is one of the most important issues of ALS data processing. In this paper, the authors present a previously unknown, yet simple and efficient method for altimetric enhancement of ALS data based on the concept of lidargrammetry. The generally known photogrammetric theory of stereo model deformations caused by relative orientation parameters errors of stereopair was applied for the continuous correction of lidar data based on ground control points. The preliminary findings suggest that the method is correct, efficient and precise, whilst the correction of the point cloud is continuous. The theory of the method and its implementation within the research software are presented in the text. Several tests were performed on synthetic and real data. The most significant results are presented and discussed in the article together with a discussion of the potential of lidargrammetry, and the main directions of future research are also mapped out. These results confirm that the research gap in the area of altimetric enhancement of ALS data without additional trajectory data is resolved in this study.

**Keywords:** UAV-ALS data; data enhancement; lidargrammetry; altimetric correction

## 1. Introduction

Since aerial laser scanning (ALS) became a common mapping technology, the problem of its accuracy has been discussed and investigated [1,2]. The ALS data accuracy is concerned with the accuracy of raw scanning effect [3,4] or the accuracy of the digital elevation model (DEM) [5,6]. An accuracy assessment of the ALS data is usually based on ground control points (GCPs) [7], intersection of the lines [8], lines (or edges) [9], planes [10] or spatial objects, e.g., boxes [11]. Several authors describe the sources of systematic and random errors of ALS data [12–15]. There are more accuracy factors of ALS data than in photogrammetric mapping [1].

### 1.1. ALS Data Accuracy Assessment

More complex and advanced solutions for the assessment and enhancement of accuracy are proposed on the basis of specifically designed GCPs [16], linear features [17], roofs [18] or other geometric features [19]. The algorithms use the methods known from photogrammetric block adjustment of independent models [20], iterative closest point (ICP) [13], high accuracy time interval measurement methods of pulse of LiDAR (Light Detecting and Ranging) [15] and the calculation models for temporal variance and spatial variance for taking into account the physical conditions of the atmosphere [21]. Recently, LiDAR platforms for unmanned aerial vehicles (UAVs) have gained increasing popularity, whilst their accuracy is assessed on test fields [22] for digital terrain model (DTM) generation [23] or for specific purposes like forest analysis [24–27].

There are several commercial solutions for ALS data registration. The most popular comprise TerraMatch software of Terrasolid Ltd. [28] and Riegl's RiProcess software [29].

There are two main options for data registration implemented in TerraMatch software: surface-to-surface matching and tie lines matching, according to the type of matching method. The surface-to-surface method has three main steps: finding matches for heading, roll and pitch angles and mirror scale corrections for the whole dataset; finding matches for elevation correction or roll and elevation correction for individual strips; and finally, finding so-called fluctuations, which is elevation correction for overlapping strips in short, time-parametrised strip sections. Solvable parameters of this method comprise easting, northing and elevation shifts and drifts, heading, roll and pitch shifts and drifts, mirror scale and fluctuating elevation.

The tie lines matching analogically have these three steps, but the reference data are basically roofs' intersecting lines and other geometrical edges. In this case, the solvable parameters are:

- Easting, northing and elevation shifts;
- Heading, roll and pitch shifts;
- Fluctuating easting/northing, elevation, and roll and pitch.

All these processes need trajectory data for the calculation.

RiProcess software has three main options for strips adjustment. The first is "no rigid no translation", which is the trajectory correction without easting, northing and elevation shifts of all the data (global shifts). The second option, "no rigid with translation", allows to correct the trajectory with global shifts. The last option ("rigid with translation") translates the data globally without any local shifts and drifts of the trajectory.

In both cases—i.e., the Riegl software and Terrasolid software—the trajectory data is necessary because the method is based on the main principle of ALS scanning: each point of the point cloud is a function of adequate trajectory point, measured-out angle and measured distance. The point cloud modifications are a consequence of trajectory changes. All the above methods could be considered trajectory data-driven methods. The approach presented here does not require the trajectory data. The point cloud is directly modified, and therefore it can be named as a point data-driven method, whilst the trajectory could be modified as a secondary matter.

### 1.2. Lidar and Image Data Integration

There are three main possible kinds of analytical workflow for ALS enhancement by integration with image data [30]: so-called co-registration, which is a common and simultaneous adjustment of external orientation parameter (EOP) and point cloud position [31–33]; using lidar data for image EOP adjustment [34–36]; and using image data for system calibration [37] or registration [38,39]. The approaches of lidar data with image data registration and enhancement can be divided into two groups of methods: rigid methods and non-rigid methods [40]. The rigid methods are categorized as system-driven methods and data-driven methods. System-driven methods are based on calibration of the system: scanners and camera and additional sensors. It can be processed with the use of feature extraction [41], special targets [42] and by direct georeferencing of lidar data and images together, if calibration parameters of separate sensors and georeferencing are available [43].

Data-driven methods, based on the features extracted from point clouds and images, can be divided further into intensity-based methods and feature-based methods. Intensity-based methods use the similarity between images and intensity factor of point clouds, and based on these relations, they lead to better registration accuracy [44]. Feature-based methods use features found on both sets of data qualified as the corresponding ones [45,46].

The second group of methods comprises non-rigid methods: the methods of rigid registration with non-rigid correction and the methods of piecewise rigid registration. Rigid registration with non-rigid correction methods reduces the influence of internal calibration errors of the scanner or other nonlinearity errors of the devices. First, the ICP is used for robust registration, and then the lidar data is refined by thin-plate splines [47] or cubic splines [48]. Motion distortion of single scans before registration also has to be corrected, which can be done successfully [49]. The piecewise rigid registration is applied mostly to

mobile mapping lidar data, where the trajectory is divided into short segments, which are treated as rigid constituents of non-rigid totality [50,51]. Originally, the method was used for medical applications [52] and also for precise registration of images of a rolling shutter camera [53].

Generally, within the last few years, the proposed solutions have become more and more mathematically and physically complex. Park et al. presented a solution based on a deep convolutional neural network (CNN) and the fusion of LiDAR data and dense depth information of stereo images [54]. Nguyen et al. determined the relative position of both datasets by graph transformation matching (GTM) of a 3D building segmented from lidar data and 2D building segments of image data [55]. Li et al. first used Structure from Motion (SfM) (IMU- and GNSS-aided) of UAV images and then iteratively minimised the differences between the depth maps derived from SfM and the raw lidar data [40]. Some researchers convert lidar data to lidar intensity images [56,57] or lidar intensity and elevation images [58] for further data integration processing.

### 1.3. Lidargrammetry

The idea of discrete lidar data conversion to raster data and the usage of synthetic image with optical images is very close to lidargrammetry. The generation and application of synthetic lidar images for different purposes are present in the literature worldwide since the late 1990s [59–61]. The term "lidargrammetry" is used [62] for the comparison of different methods of fusion of "airborne laser scanning-imagery". The definition specifies that: "lidargrammetry concerns the production of inferred stereopairs (ISPs) from LiDAR intensity images, intended to stereo digitize spatial data in digital photogrammetric stations" [63], and the authors refer to the experience and solutions of the Geocue Group [64–66], which developed the algorithm and software for lidar data stereo plotting. At that time, some other companies, such as MD Atlantic and NIIRS10 [67], Optimal Geomatics [68] and Dephos Software Ltd. [69], also used and developed lidargrammetry in their technology. Analogically, the stereoscopy was applied on the basis of lidar data but also from SAR data (stereo radargrammetry) for digital surface model (DSM) generation [70].

There are two basic approaches to lidargrammetry [63]. The first approach consists in the generation of intensity lidar orthophotos with stereo-matches, based on photogrammetric traditional theory of stereo-orthophotos, whilst the [71–75] second approach is supposed to generate stereopairs of central projection images [69,76].

Teo et al. generate not only synthetic stereo images of intensity, but also range images [76]. Geocue patent is based on the generation of two tiled overlapping triangulated irregular networks (TOTIN), and stereo images are generated directly from these TOTINs [66]. The synthetic stereopairs can also be generated in a slopeward direction (e.g., for better building footprint visibility), and the precision of the stereo plotting can be appropriately modelled by B/H (base to height) ratio, and flexibly adopted to different conditions according to on-line accuracy analysis results [77]. Additionally, the point clouds are being densified: additional points are added in obscured areas with the use of Delaunay triangulation [78].

There are many different objectives of lidargrammetry application. Stereo observation of the virtual model of synthetic images is the most obvious and common for most researchers of lidargrammetry. The model can be used for stereo plotting and feature extraction [77,79], building footprint measurements [63], lidar data quality control [79] or data registration [59].

### 1.4. The Objective of the Research

In this research, the authors used the idea of conversion of discrete lidar data to synthetic images with arbitrary external and internal orientation parameters (EOP and IOP) for altimetric data enhancement. All of the proposed methods of lidar data enhancement, based on integration with image data, require both sets of data: lidar and photogrammetric data. Contemporary methods of enhancement of lidar data applying approaches other

than integration with photogrammetric data are also complex and sophisticated. A typical approach is a trajectory data-driven method.

In this paper, a solution is proposed in the form of the lidargrammetric model deformation method (LMD), which consists of a simple and effective approach for altimetric ALS data enhancement. The theory of photogrammetric model deformation caused by errors of parameters of relative orientation was used. This approach is simple and does not require the trajectory data. That is why this can be defined as a point data-driven method.

## 2. Materials and Methods

This approach is dedicated to a block of ALS data, divided into single strips that are not precisely consistent in height, whilst the solution might be applied in situations where:

- The data are postprocessed and their trajectory data are not accessible;
- It is not possible to enhance the data consistency with the assessable trajectory data applications.

There are two cases in which height discrepancies of two overlapping ALS strips or a single ALS strip are known:

- As height differences of two ALS strips measured in an overlapping area;
- As a height difference between strip side borders and ground control points (GCPs) or existing DTM/DSM.

The strip deformation can be defined thanks to the height differences. Points of height are located symmetrically to the approximated flight line projected on the XY plane of the global coordinate system, close to the side borders of the strip. To simplify the terminology, the pairs of these points are named as "point profile–PP". The first PP should be located at the beginning of the strip whilst the last one should be at the end, and the distribution of intermediate PP should guarantee appropriate correction of the strip. The strip is divided into segments by the intermediate PP. Further details will be provided by description of the method. The corrections according to the two cases can be calculated in two manners.

In the first case, the correction is a half of the difference between the overlapping strips, and such a correction will be applied with a positive sign to the strip which is lower and with a negative sign to the upper strip.

In the second case, the correction is equal to the control point height minus strip height and is applied with a positive sign.

The theory of photogrammetry describes the problem of height deformations of a stereoscopic model caused by errors of relative orientation parameters (ROP) [80]. Figure 1 presents a general overview of this problem.

The height deformation of the stereo model presented in Figure 1 is caused by ROPs and can be described according to K. Kraus [80] by following a mathematical relation of the height model deformation:

$$dZ = dZ_{12} - \frac{(X - B)}{B}dB_z + \frac{XY}{B}dom - \frac{YH}{B}dka \tag{1}$$

where $dZ$ is a height model deformation in the model point ($X$, $Y$, $H$ coordinates), $B$ is a photo base (distance between the centres of projection), $dB_Z$ is a height error of the first projection centre, $dom$–error of $\omega$ angle of the first photo, $dka$–error of $\kappa$ angle of the first photo. The Equation (1) has to be converted in order to calculate ROP corrections ($dB_Z$, $dom$, $dka$) knowing the height deformation in the strip corners, represented by GCPs defined by XYH coordinates. The height shift of both projection centres has to be done according to the average height difference of all GCPs $dZ_{12}$. The whole process has to be repeated iteratively because the Equation (1) describes an approximated relation between height model deformations and ROP errors.

It has been observed that one section of the ALS strip (between the next two PPs) can be considered analogically to the virtual stereo model.

In photogrammetry, stereopair implicates the virtual stereo model; in lidargrammetry, the ALS segment can implicate two lidargramms: synthetic images of a point cloud captured by a virtual camera or the nominal interior orientation parameters. The procedure is implemented in the research tool named PyLiGram (Python LidarGrammetry). Its interface is shown in Figure 2.

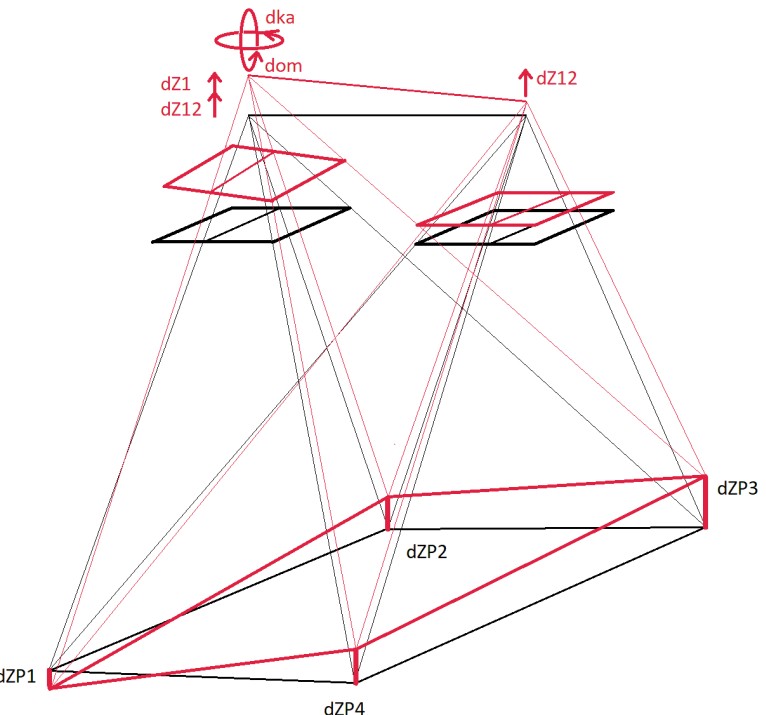

**Figure 1.** Stereoscopic model of height deformations from dZP1 to dZP4 caused by the errors of relative orientation parameters (ROP).

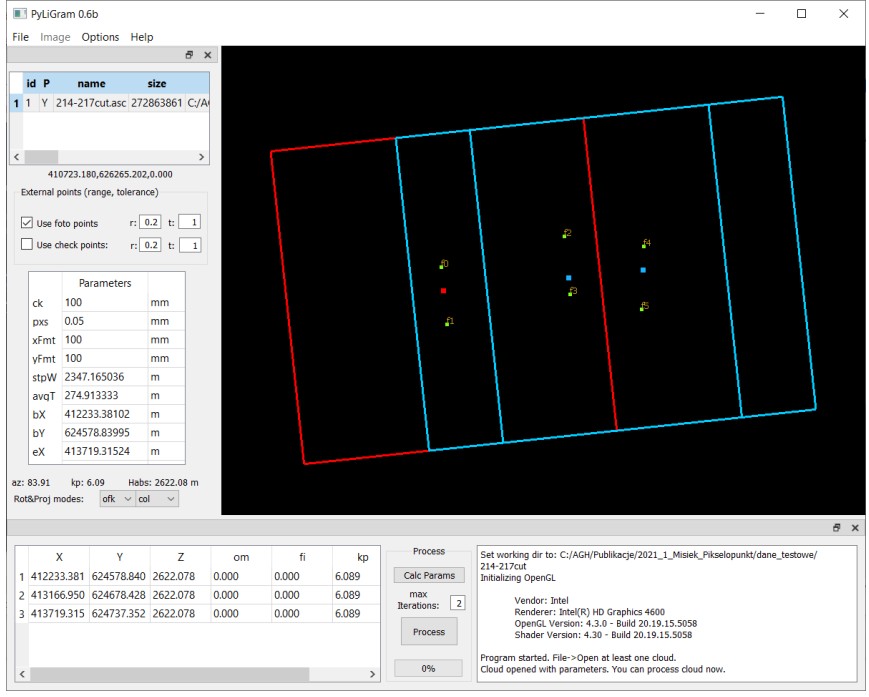

**Figure 2.** Interface of the PyLiGram research tool with ALS single strip with 6 GCPs (green) and 3 generated lidargramms.

The pair of lidargramms is a stereogram of normal, vertical images (omega = 0, phi = 0). XY coordinates of each projection centre are calculated as a point of the intersection between a regression line fitted with first degree polynomial to PP points and lines connecting pairs of PP points. Then, flight height $H_{flight}$ is calculated according to Equation (2):

$$H_{flight} = \frac{2.5 \ B \ c_k}{x_{format}}$$

(2)

where $B$ is a photo base, $x_{format}$ is the size of the lidargram in flight direction in [mm] and $c_k$ is a focal length, also in [mm]. The kappa angle depends on the virtual flight direction calculated as a straight line connecting the first and the last projection centre. The intensity value or RGB values of the points are centrally projected on the virtual image planes using collinearity equations [80]. The RGB colour of lidargram pixels is interpolated.

After the generation of lidargrams, the continuous correction of the ALS segment can be applied by changes of ROP of these lidargrams. Changes of ROPs of lidargrams lead to ALS segment (and, in consequence, to ALS strip) deformation, analogically to errors of the ROPs of the photos which cause deformation of the stereo model.

The discrepancies of the four corners of the ALS segment can be corrected in four basic steps.

- The first step is to change the height of both projection centres of the lidargrams dZ12 (Figure 3a).
- The second step is to change the height of the centre of the left lidargram dZ1 (Figure 3b).
- The third step is to change the kappa angle of the left lidargram dka (Figure 3c).
- The fourth step is to change the omega angle of the left lidargram dom (Figure 3d).

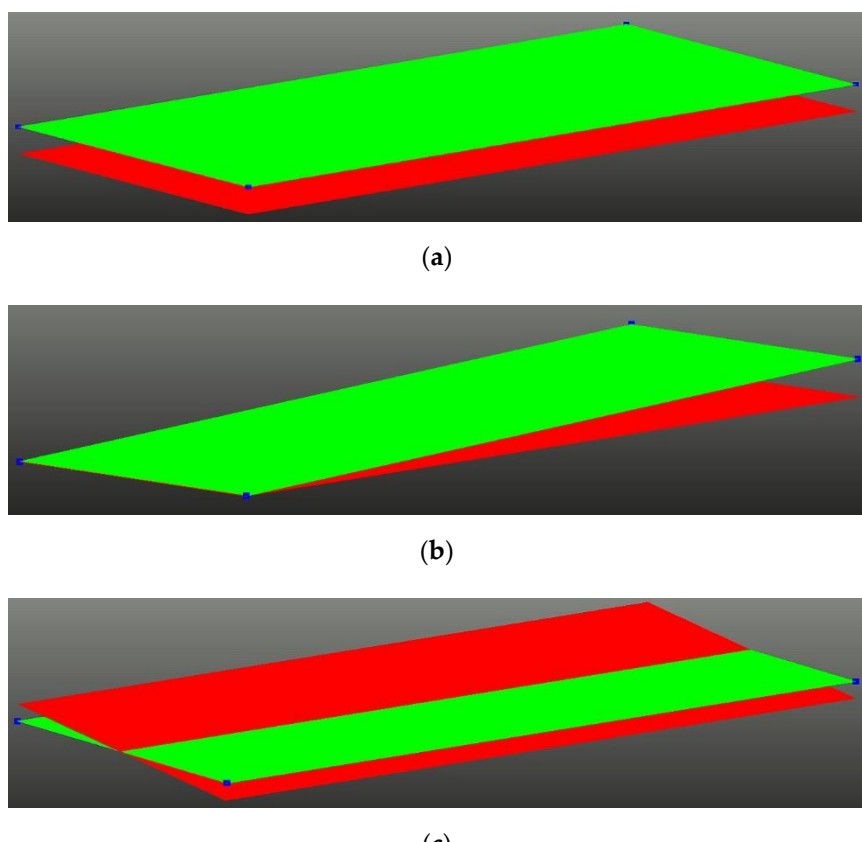

(**a**)

(**b**)

(**c**)

**Figure 3.** *Cont.*

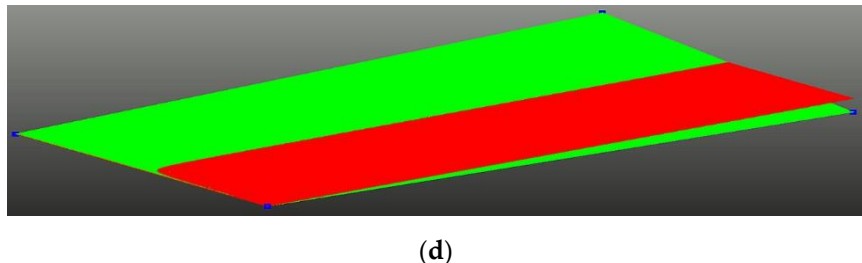

(**d**)

**Figure 3.** Synthetic point cloud before (red) and after correction (green). Blue dots are the GCPs. The whole process is shown separately by four basic steps: (**a**) dZ12; (**b**) dZ1; (**c**) dkappa; (**d**) domega.

The combination of these steps is used as corrections of the EOP when the corrected point cloud is going to be generated by space intersection. For the first segment of the ALS strip, the left lidargram is rotated by dkappa and domega, and its projection centre height Z is changed by dZ12 and dZ1. The right lidargram centre of projection is only changed by dZ12. Using such a new, corrected EOP, the new point cloud is calculated by photogrammetric intersection.

The method can be used for single-strip processing and for an entire block of strip processing. The general organization of the ALS data correction process is presented in Figure 4.

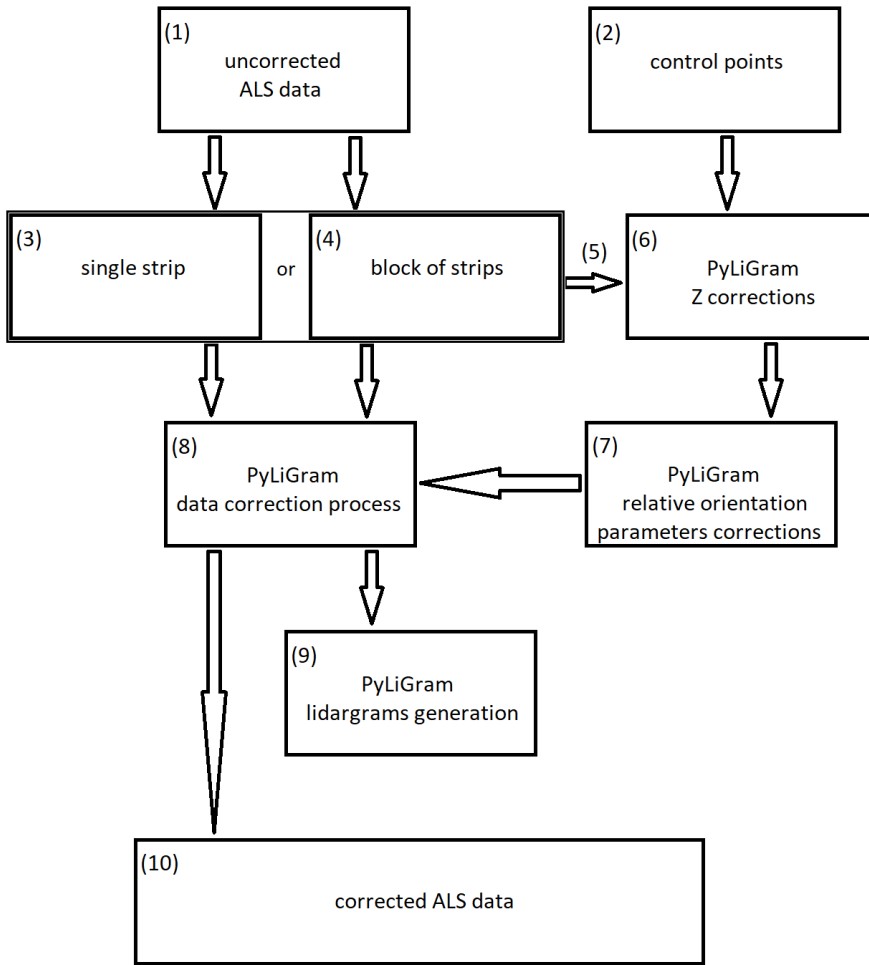

**Figure 4.** The general workflow of ALS data height correction. Detailed explanation in the text below.

The input is ALS uncorrected data (1) as single strips (3) or block of strips (4) and GCPs (2) distributed as is shown in Figure 5. The Z corrections (6) can be calculated by means of

comparison of height of GCPs and point cloud or by means of comparison of the height of two overlapping ALS strips (5). The height of the point cloud at a specific XY point is calculated as an average height of this XY point surrounding taken into account: inside of a circle and in height tolerance. In the next step, the corrections of relative orientation parameters (ROPs) (7) are calculated based on Z corrections according to the converted Equation (1). The strips of ALS data are divided into segments and processed separately within segments. The first pair of GCPs defines the beginning of the segment, whilst the second pair of GCPs defines its end and the beginning of the next segment. Figure 5 presents one strip with 6 GCPs divided into 2 segments. The data-correction process (8) is a calculation of the new corrected point cloud relying on the spatial intersection of the homologous rays. The algorithm uses collinearity equations and external orientation parameters corrected by ROPs' corrections. The new point cloud is generated in segments (10).

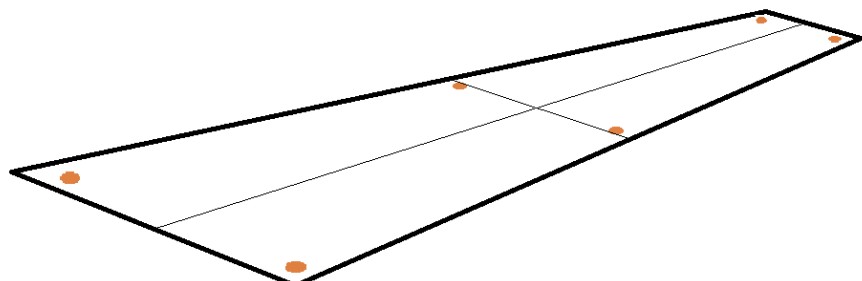

**Figure 5.** An example of the distribution of six GCPs within one ALS data strip divided into two segments.

After the first phase of research, the optimization change was applied. The ALS data correction based on stereo deformation model theory is done without generating and saving the pair of lidargrams on the disk. The process consists in correcting point by point and the lidargrams become only additional virtual objects and not crucial to the process. They can be generated optionally (9).

## 3. Results and Discussion

The presented method was tested on synthetic (a-tests) and real datasets (b-tests). The were three basic tests: test 1—the correction of the single segment (short strip with four GCPs); test 2—test of joints of two corrected segments; and test 3—test of correspondence of two parallel overlapping strips.

### 3.1. Test 1a. Single Segment Correction Test of Synthetic Data

The flat synthetic strip was generated by the additional Python tool. All the points were in a regular 0.1 m grid and of the same height. The GCPs were situated in the corners moved slightly away from the edge of the point cloud. Their heights should cause all four ROP corrections. The distribution of four GCPs and 21 check points is shown in Figure 6. The check points were distributed on sloped straight lines to check whether the transversal and longitudinal profiles of the corrected point cloud would be straight.

The result of the correction was reported and the height deviations on GCPs were reduced to 0.000 m after the second iteration (Table 1), and the check point differences were less than 0.001 m (point cloud averaged height). The presented result shows that the corrections of the ROPs were properly.calculated and applied. The process was successful, and the transversal and longitudinal profiles of such a corrected point cloud are straight lines. In the other case, the slope profiles (neither transversal nor longitudinal) are not straight in general.

### 3.2. Test 1b. Single Segments Correction Test of Real Data

The test data were point cloud of Riegl VUX-1UAV scanning system (Figure 7). A local 500-m-long road in the Krakow area was scanned. GCPs and check points were prepared

by selection of data location and manual entry of height discrepancies. The average point distance between 11.1 million points was about 0.05 m.

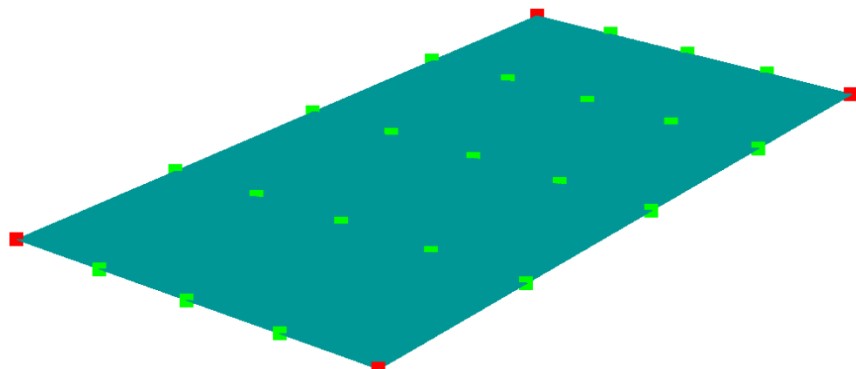

**Figure 6.** The synthetic data with four GCPs (red) and 21 check points (green).

**Table 1.** Height discrepancies of single segment of synthetic data before, during and after the process and ROP corrections of the lidargram model.

| Height Discrep. | ftp1 [m] | ftp2 [m] | ftp3 [m] | ftp4 [m] |
|---|---|---|---|---|
| Before process | −0.100 | 0.400 | 0.200 | −0.300 |
| After 1st iteration | 0.000 | 0.000 | −0.028 | −0.028 |
| Final result | 0.000 | 0.000 | 0.000 | 0.000 |
| **ROP corrections:** | **dZ12 [m]** | **dZ1 [m]** | **dka [deg]** | **dom [deg]** |
| | 0.150 | −0.228 | −0.00201 | −0.01005 |

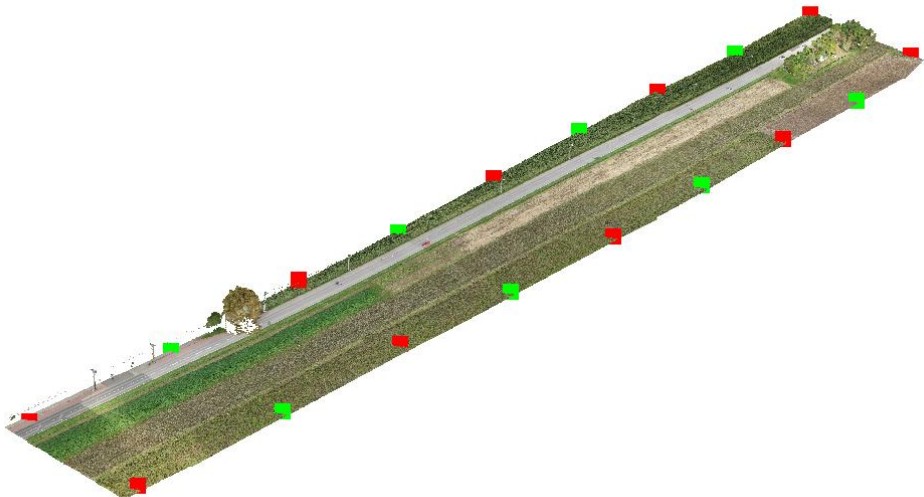

**Figure 7.** Strip of UAV lidar data with ten GCPs (red) and eight check points (green).

The data were corrected in four segments, and the results were analogical to the results of the synthetic data: the differences on each GCP were 0.000 m after the third iteration (Table 2), and the ROP corrections were calculated and applied properly.

The height differences of check points were below standard deviations of height surrounding the check point (Table 3).

### 3.3. Test 2a. Segments' Joint Testing of Synthetic Data

The next test was performed for the joints of the segments in the case of nonregular GCP distribution. The synthetic data were corrected based on GCPs but two central GCPs were shifted along a flight direction; one 10 m forward, and the other 10 m backward. For a

strip width of 100 m, the height corrections defined by GCPs were about ±0.3 m. After the correction process, two segments were created as shown in Figure 8.

**Table 2.** Height discrepancies of four segments of real data before, during and after the process and final values of ROP corrections.

| Height Discrep. | # of Segment | ftp1 [m] | ftp2 [m] | ftp3 [m] | ftp4 [m] | ftp5 [m] | ftp6 [m] | ftp7 [m] | ftp8 [m] | ftp9 [m] | ftp10 [m] |
|---|---|---|---|---|---|---|---|---|---|---|---|
| Before process | 1 | 0.2 | 0.61 | 0.1 | 0.894 | - | - | - | - | - | - |
| | 2 | - | - | 0.1 | 0.894 | 0.6 | 0.69 | - | - | - | - |
| | 3 | - | - | - | - | 0.6 | 0.69 | 0.681 | 0.753 | - | - |
| | 4 | - | - | - | - | - | - | 0.681 | 0.753 | 0.789 | 0.637 |
| After 1st iteration | 1 | 0.001 | −0.0 | −0.003 | −0.009 | - | - | - | - | | |
| | 2 | - | - | 0.009 | −0.021 | −0.012 | −0.072 | - | - | | |
| | 3 | - | - | - | - | 0.013 | −0.016 | 0.005 | −0.005 | | |
| | 4 | - | - | - | - | - | - | 0.0 | 0.002 | 0.001 | −0.005 |
| After 2nd iteration | 1 | 0.000 | 0.000 | 0.000 | 0.000 | - | - | - | - | - | - |
| | 2 | - | - | −0.003 | 0.003 | −0.006 | 0.002 | - | - | - | - |
| | 3 | - | - | - | - | 0.000 | 0.000 | 0.000 | 0.000 | - | - |
| | 4 | - | - | - | - | - | - | 0.000 | 0.000 | 0.001 | −0.001 |
| After 3rd iteration (final result) | 1 | 0.000 | 0.000 | 0.000 | 0.000 | - | - | - | - | - | - |
| | 2 | - | - | 0.000 | 0.000 | 0.000 | 0.000 | - | - | - | - |
| | 3 | - | - | - | - | 0.000 | 0.000 | 0.000 | 0.000 | - | - |
| | 4 | - | - | - | - | - | - | 0.000 | 0.000 | 0.000 | 0.000 |

| ROP corrections: | # of segment | $dZ_{i(i+1)}$ [m] | $dZ_i$ [m] | dka [deg] | dom [deg] |
|---|---|---|---|---|---|
| | 1 | 0.427 | 0.038 | 0.00302 | −0.00755 |
| | 2 | 0.465 | 0.134 | 0.00587 | 0.01389 |
| | 3 | 0.640 | 0.076 | 0.00054 | 0.00039 |
| | 4 | 0.717 | −0.013 | 0.00041 | 0.00363 |

**Table 3.** Height discrepancies of check points and the standard deviations of the height of the surrounding check point.

| | chp1 [m] | chp2 [m] | chp3 [m] | chp4 [m] | chp5 [m] | chp6 [m] | chp7 [m] | chp8 [m] |
|---|---|---|---|---|---|---|---|---|
| Height discrep. | 0.003 | 0.005 | −0.001 | −0.001 | −0.005 | 0.004 | 0.003 | −0.003 |
| Height std. dev. | 0.006 | 0.095 | 0.357 | 0.074 | 0.271 | 0.352 | 0.456 | 0.031 |

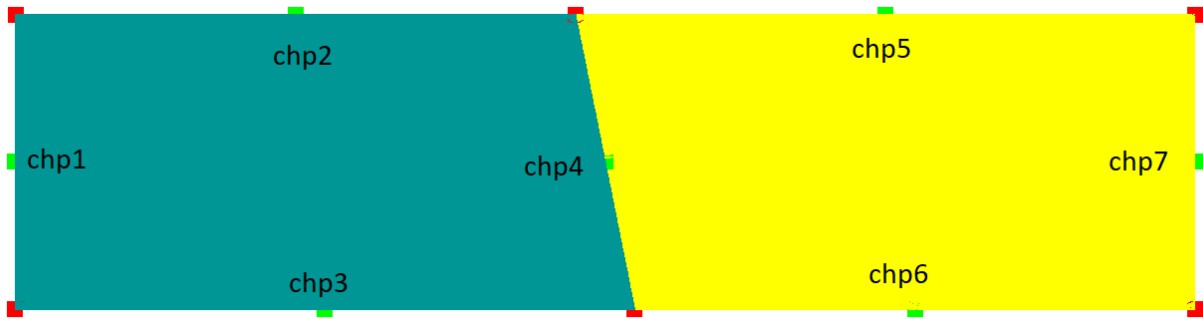

**Figure 8.** Top view of two segments generated after correction process with red GCPs and green check points distribution.

The result was positive but a significant disadvantage of this method was observed; in the joints of the segments, an aperture height dZ in extreme check point nr 4 of 0.052 m was observed (Table 4 and Figure 9).

**Table 4.** Height discrepancies of check points with 0.052 m aperture at chp4.

| Height Discrep. | # of Segment | chp1 [m] | chp2 [m] | chp3 [m] | chp4 [m] | chp5 [m] | chp6 [m] | chp7 [m] |
|---|---|---|---|---|---|---|---|---|
| | 1 | −0.0 | 0.001 | 0.001 | **−0.025** | | | |
| | 2 | | | | **0.027** | 0.001 | 0.0 | 0.001 |

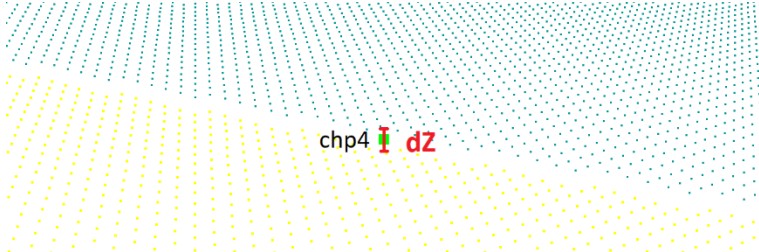

**Figure 9.** Magnification of the aperture dZ between two segments.

The reason for this problem is clarified in a description of test 1a: the transversal and longitudinal profiles are straight, whereas all the others are not. Two central GCPs do not lie symmetrically to the central line. They define the segment joints as neither transversal nor longitudinal; they are sloped to the flight line. Only transversal and longitudinal profiles of the deformed point cloud are straight. In this case, the yellow segment along this profile is concave and the blue one is convex. The probable solution to this problem, which will be implemented in the next release of the PyLiGram tool, is to align the joints of the segments based on the average weight of the calculated heights of the points in the transition area. The aperture is not significant, but the problem has to be solved. Meanwhile, the GCPs have to be distributed as symmetrically to the flight line as possible.

### 3.4. Test 2b. Segments' Joint Testing of Real Data

The real data were also captured by the UAV Reigl system with a VUX-1UAV scanner. This specific fragment of S1 highway in Silesia was selected because of its curvature (Figure 10). As before, the GCPs and check points were selected arbitrarily, and the altitude discrepancies were applied. The average point distance between 11 million cloud points was about 0.15 m.

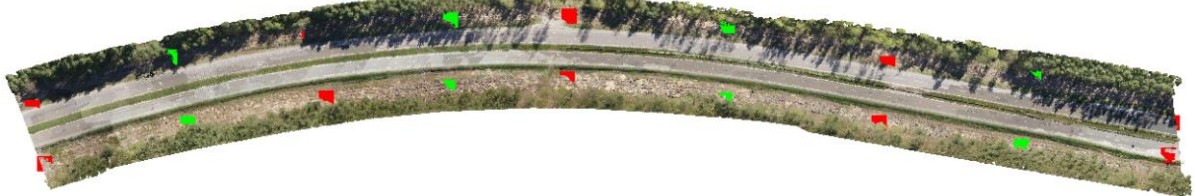

**Figure 10.** The curved highway with four segments and irregular GCP distribution.

The curvature and forest caused an irregular GCP distribution. The correction in this case did not cause significant apertures between segments (Figure 11).

### 3.5. Test 3a. Testing of Correspondence of Two Parallel Overlapping Strips of Synthetic Data

The last two tests were dedicated to the problem of correspondence of the overlapping strips (Figure 12). The first test was done on synthetic data: two parallel and overlapping strips of different height. The strips were corrected by six GCPs each, three of which were common, and the correctness of correspondence of the strips was checked by two points located in the overlapping area.

The corrections defined on GCPs were about ±0.3 m. After the correction process, the differences between the checkpoints and corrected strips were similar and came to about 0.001–0.002 m (Table 5).

**Table 5.** Height discrepancies of check points between strips of synthetic data.

| Height Discrep. | Strip | chp1 [m] | chp2 [m] |
|---|---|---|---|
| | 1 | 0.001 | 0.001 |
| | 2 | 0.001 | 0.002 |

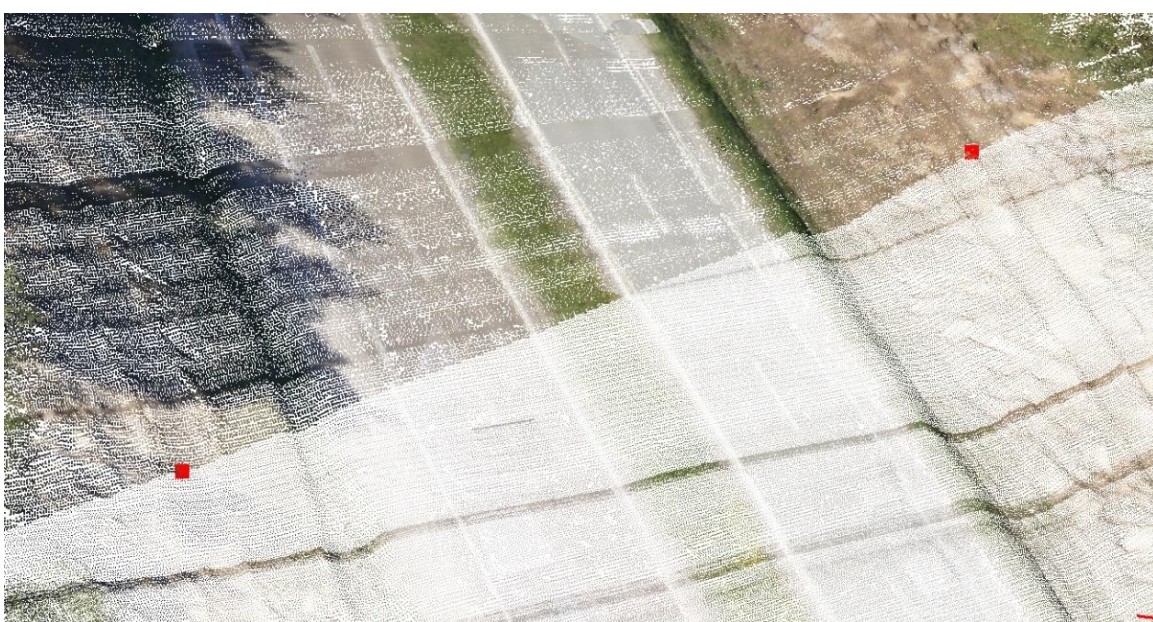

**Figure 11.** A correct joint of segments of the highway test case with ref GCPs. The joint is seen thanks to different cloud point sizes.

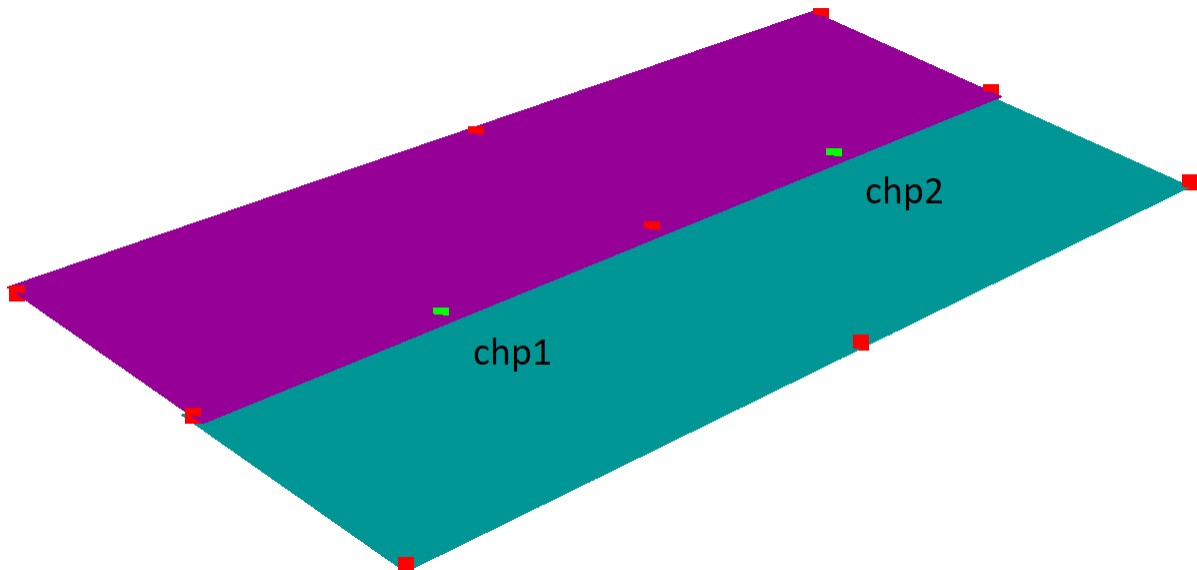

**Figure 12.** Two synthetic parallel strips with nine red GCPs and two green check points.

### 3.6. Test 3b. Testing of Correspondence of Two Parallel Overlapping Strips of Real Data

The last test field was an ALS data of Krakow's centre. One block of real archival data from 2010 was chosen, and two overlapping strips were extracted afterwards (strip one of 4.6 million points, strip two of 4.7 million points). The average distance of points was 0.20 m. There were six GCPs measured on this data (two in the overlapping area), and there were three check points also chosen in the overlapping area (Figure 13). The heights of GCPs were changed: the real data were first deformed by these GCPs with changed height. Then, both strips were corrected back using the GCPs with original heights. The correspondence of the strips was checked on three check points (Table 6). The result was positive within accuracy of height calculation from cloud points within selected radius and tolerance. After two to three iterations of deforming and then two to three iterations of correcting, both processes (one after the other) gave the initial position of the data. This means that the process works correctly and precisely.

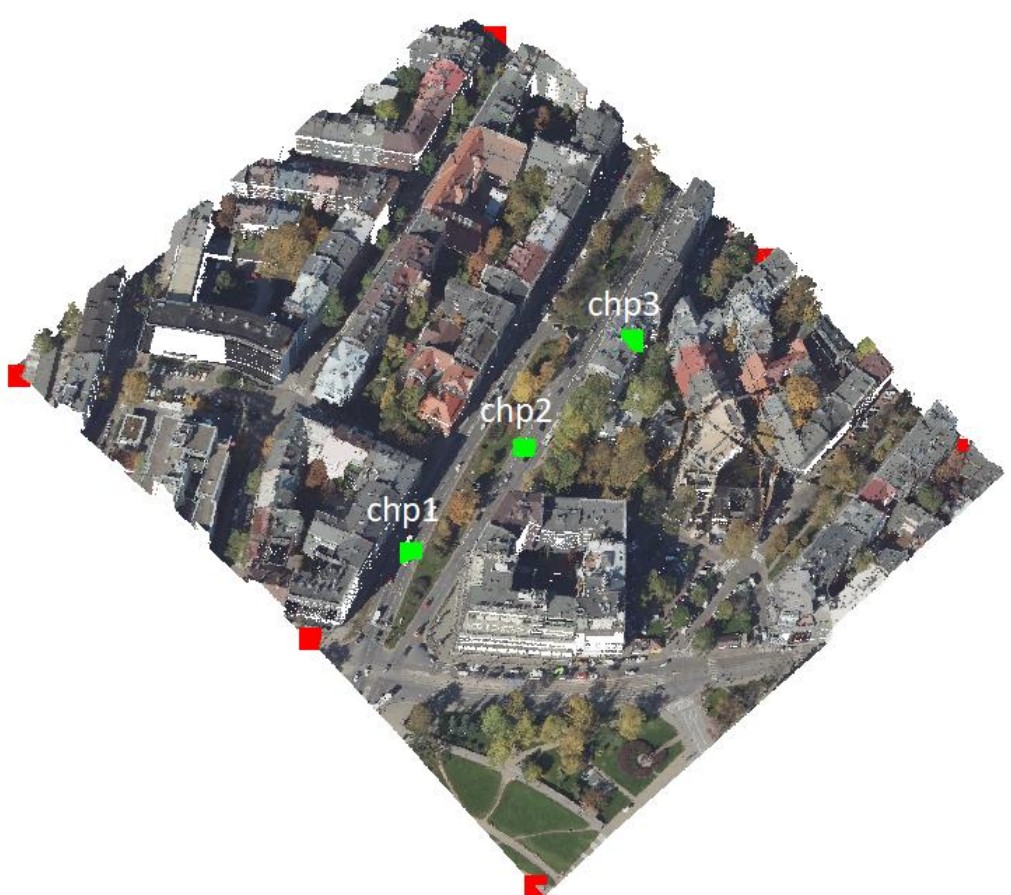

**Figure 13.** Two strips of real data of Krakow centre, with six red GCPs and three green check points.

**Table 6.** Height discrepancies of check points between strips of real data.

| Height Discrep. | Strip | chp1 [m] | chp2 [m] | chp3 [m] |
|---|---|---|---|---|
| | 1 | 0.015 | 0.015 | 0.000 |
| | 2 | 0.015 | 0.015 | 0.000 |

### 3.7. Summary of the Tests

All of the six presented tests are selected from a large amount of method tests and tests performed on the PyLiGram application to present the possibilities of the method; its potential and its limitations.

The synthetic data tests were very important and gave the first indicators of the correctness of the method. It is significant that the greatest disadvantage of the method was observed and specified on synthetic (rather than real) data. The synthetic data initially had the same height, so possible unexpected results could not be interpreted as a result of rough point cloud height calculation. The tests on real data were processed to check the method with regards to the data with noise, low and medium vegetation, buildings and a real situation: a straight road going through fields, a highway in the forest and an area of the city centre.

The general results are: the method is correct, efficient and precise. The correction of the point cloud is continuous. The thesis of the research, stating that the lidargrammetric model deformation method (LMD) makes up a simple and effective approach for altimetric ALS data enhancement, is confirmed by the results of the tests on synthetic and real data. The only limitation of our approach is the need for GCPs in correct distribution.

The LMD method is applicable not only to UAV, but to all kinds of aerial LiDAR data, including typical ALS point clouds captured by manned aircrafts.

## 4. Conclusions

The LMD method (implemented in the research software PyLiGram) presented here is an optional, new, effective and precise method of ALS data enhancement using integration of LiDAR data processing with a photogrammetric approach by harnessing the idea of lidargrammetry. The research will be continued in several directions, and the PyLiGram tools will be further developed. There is probably a big potential in this method, and also in lidargrammetry in general. The additional correction by phi angle application will be implemented in the case of more GCP data. Automatic methods of strips overlapping area alignment will be the subject of further research. The authors of this study plan to solve the problem of aperture in segment joints by weighted average height calculations and to change the means of calculation of the strip central line method as well. The segments will have their own central lines to make the method more flexible for irregular strips of UAV scanning systems. The biggest limitation of the method is the need for the GCPs, currently with regular distribution.

The innovation and application value of the LMD method compared with typical processing workflows is that simplicity of the algorithm and possibility to enhance the altimetric data accuracy does not need the trajectory data. The method can be applied to all lidar datasets and also to archival datasets. The only need is for GCPs and their appropriate distribution within LiDAR strips. In the case of archival LiDAR blocks, it is necessary to cut it in strips according to the location of GCPs. Such strips ought to have overlapping areas where the GCPs are. The LMD method is flexible and adaptable to a large amount of points—LiDAR data are usually very big. The process splits points into smaller chunks, so the algorithm does not require large memory resources or high computing power.

The LMD method is developed in the wider context of using lidargrammetric data for several purposes: seeking the optimal solution for integrated photogrammetric and point cloud data formats and using lidargrammetric data as raster and vector data for spatial edges detection. Methods of ALS data enhancement based on dense matching of lidargrams will be the main direction of the research in the near future.

**Author Contributions:** Conceptualization, A.R. and M.T.; methodology, A.R. and M.T.; software, A.R. and M.T.; validation, A.R. and M.T.; formal analysis, A.R. and M.T.; investigation, A.R. and M.T.; resources, A.R. and M.T.; data curation, A.R. and M.T.; writing—original draft preparation, A.R. and M.T.; writing—review and editing, A.R. and M.T.; visualization, A.R. and M.T.; supervision, A.R. and M.T.; project administration, A.R. and M.T. All authors have read and agreed to the published version of the manuscript.

**Funding:** This study was carried out with financial support of AGH grant No.: 16.16.150.545., "Spatial engineering, photogrammetry and remote sensing for the needs of science, economy and administration", and AGH IDUB project, "Integration of remote sensing data for control in the system of direct agricultural subsidies (IACS)".

**Data Availability Statement:** Not applicable.

**Acknowledgments:** The authors are grateful to Geodimex S.A. for their support in obtaining the data for this research.

**Conflicts of Interest:** The authors declare no conflict of interest.

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
