# Peer review of "The Lidargrammetric Model Deformation Method for Altimetric UAV-ALS Data Enhancement"

_remotesensing, doi:10.3390/rs14246391_

Round 1

Reviewer 1 Report

Suggestion for remotesensing-2000262

Authors present a simple and efficient method for altimetric enhancement of ALS data based on the concept of lidargrammetry. Overall, this study is contributive with an appropriate workflow. However, the manuscript is poor written and language needs polish by native speakers. I suggest this manuscript should be under a major revision before the further process. Some tips are in the manuscript, and the specific suggestions are follows:

1.     Abstract: The first sentence should be the importance of altimetric enhancement of ALS data. The research gap which you want to solve in this study should be reported in the second sentence. And the last sentence should be the explicit contribution of this study, which is also the solution degree of the research gap that you reported in the second sentence. The content should be more informative, the existing style of writing is just like a lab report not an academic paper. The results should be reported with crucial values of accuracy assessments and improvements compared to related papers. The innovation and explicit contribution of this study, should be emphasized in the Abstract. I think authors should adjust structure, and rewrite the content of Abstract.

2.     Keywords: The keywords should also emphasize the innovation and explicit contribution. The existing keywords showed that this research is lack of innovation and contribution.

3.     Main body: The content should be more informative, the existing style of writing is just like a lab report not an academic paper.

1)      Study area should be described.

2)      Methods: Lack of references.  

3)      Discussion: Should be listed separately from results. The uncertainty should be discussed detailed. And the results comparison with previous studies should be discussed with specific values of accuracy evaluation indices. 

Author Response

Dear Reviewer,

Thank you very much for your review. The answer is attached in PDF.

Best regards,

Authors

Author Response

(The authors gave the same response as above.)

Reviewer 3 Report

Paper focuses on method for altimetric enhancement of ALS data based on the concept of lidargrammetry. In general paper presents novel method but in my opinion requires minor revision.

My suggestions:

1. Detailed description of data sets is missing (it should be mentioned in section 2) - type of area, objects inside, device for data acquisition and its parameters.

2. Method should be described more scientifically - now it is more like lab report. Please consider making the flowchart or any other detailed schema of the method

3. Tables 1 and 2: if in ROP correction dZ values are written correctly? I think the precision is unreal. Pleasu unify the precizion of the values in tables and text.

4. Please specify the types of control and check point chosen to the test. In figures like fig. 11 or 13 it is hard to observe what kind of object was chosen.

5. Please consider the extention of the conclusions. What id the advantages of your method vs. other solutions (maybe percentage of accuraccy?) if there is any more efectiveness like degree of automation of ROP process etc.

editorial errors

It is usually not acceptable to place tables or figure one by one without any comment in teh text. Then it looks more like attachments than report.

Author Response

(The authors gave the same response as above.)

Reviewer 4 Report

The method proposed in this manuscript can correct LIDAR data without trajectory data. It is suitable for LIDAR data processing without trajectory data and it has some creative innovation.  In fact, it is the same as the photogrammetric method, which corrected by ground control points (GCPs).  I don't know whether the method is applicable with the LIDAR data obtained in case of inconsistent unmanned aerial vehicle (UAV) altitudes. I expected the authors to point out and explain this situation in the manuscript. In addition, there were still some doubts that need to be clarified:

1. I suggest that some content can be added to abstract to describe  the method and results.

2. line 73: Acronym appearing for the first time should be explained, such as EOP. 

3. line 104: I don't konw what SfM means. 

4. line 133:  What does B/H ratio mean?

5.  Why are the check points of strip of UAV lidar data distributed at the boundary in figure7. 

6. Is figure 13 from lidargrammetry? 

7. I don't know whether the trajectory data of LIDAR data is from UAV or other inertial navigation systems. 

8. Line 366-376:  How to obtian the real data in "Test 3b"? 

Author Response

(The authors gave the same response as above.)

Round 2

Reviewer 1 Report

My main concerns were solved